# Latent Imputation before Prediction:
# A New Computational Paradigm for *De Novo* Peptide Sequencing

Ye Du [1]   Chen Yang [1]   Nanxi Yu [1]   Wanyu Lin [2]   Qian Zhao [3]   Shujun Wang [1 4 5]

## Abstract

*De novo* peptide sequencing is a fundamental computational technique for ascertaining amino acid sequences of peptides directly from tandem mass spectrometry data, eliminating the need for reference databases. Cutting-edge models usually encode the observed mass spectra into latent representations from which peptides are predicted autoregressively. However, the issue of missing fragmentation, attributable to factors such as sub-optimal fragmentation efficiency and instrumental constraints, presents a formidable challenge in practical applications. To tackle this obstacle, we propose a novel computational paradigm called **L**atent **I**mputation before **P**rediction (LIPNovo). LIPNovo is devised to compensate for missing fragmentation information within observed spectra before executing the final peptide prediction. Rather than generating raw missing data, LIPNovo performs imputation in the latent space, guided by the theoretical peak profile of the target peptide sequence. The imputation process is conceptualized as a set-prediction problem, utilizing a set of learnable peak queries to reason about the relationships among observed peaks and directly generate the latent representations of theoretical peaks through optimal bipartite matching. In this way, LIPNovo manages to supplement missing information during inference and thus boosts performance. Despite its simplicity, experiments on three benchmark datasets demonstrate that LIPNovo outperforms state-of-the-art

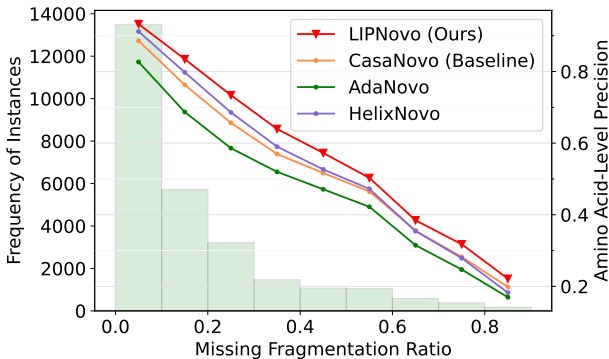

*Figure 1.* Comparison of amino acid-level precision between LIP-Novo (ours) and existing methods under varying missing fragmentation ratios. As the missing ratio increases, performance deteriorates dramatically, highlighting the detrimental impact of the missing fragmentation issue. The proposed LIPNovo consistently outperforms existing methods across all missing ratios. Results are based on the test set (*i.e.*, the yeast species) from the Nine-species dataset (Tran et al., 2017).

methods by large margins. Code is available at https://github.com/usr922/LIPNovo.

## 1. Introduction

Peptide sequencing, *i.e.*, determining the amino acid sequences of peptides from observed mass spectra, is the cornerstone of the typical tandem mass spectrometry (MS) based proteomics workflow (Aebersold & Mann, 2003; Nesvizhskii et al., 2007; Strauss et al., 2024). This technique plays a pivotal role in understanding protein structure and function (Lee et al., 2007; Nowinski et al., 2012), holding significant importance in various applications such as drug discovery (Lin et al., 2020), biomarker discovery (McDonald & Yates Iii, 2002; Wenk et al., 2024), and medical research (Uzozie & Aebersold, 2018; Macklin et al., 2020).

Early approaches (Noor et al., 2021) to solving this task rely on a *database search* paradigm, where each spectrum is scored against a set of candidate peptides, and the highest-scoring peptide-spectrum match (PSM) is retrieved as the prediction. However, this approach is susceptible to certain limitations, such as limited coverage of the peptide space in the database and difficulties in identifying new or rare

[1]Department of Biomedical Engineering, The Hong Kong Polytechnic University. [2]Department of Computing, The Hong Kong Polytechnic University. [3]Department of Applied Biology and Chemical Technology, The Hong Kong Polytechnic University. [4]Research Institute for Smart Ageing, The Hong Kong Polytechnic University. [5]Research Institute for Artificial Intelligence of Things, The Hong Kong Polytechnic University. Correspondence to: Shujun Wang <shu-jun.wang@polyu.edu.hk>.

*Proceedings of the 42nd International Conference on Machine Learning*, Vancouver, Canada. PMLR 267, 2025. Copyright 2025 by the author(s).

peptides. In response to these limitations, the field has witnessed the emergence of the *de novo* peptide sequencing paradigm, propelled by the rapid advancements of deep learning (Tran et al., 2017). Unlike traditional database-based approaches, *de novo* sequencing does not depend on prior knowledge from a pre-constructed protein database. Instead, it focuses on reconstructing peptide sequences directly from the observed spectra. This capability facilitates the identification of previously unseen peptides such as mAbs (Bandeira et al., 2008).

Current *de novo* sequencing methods (Tran et al., 2017; Qiao et al., 2021; Yilmaz et al., 2022; Eloff et al., 2023; Xia et al., 2024; Yang et al., 2024; Yilmaz et al., 2024) typically employ an encoder-decoder architecture. Within this framework, a spectrum encoder transforms the observed spectra into latent representations, which are subsequently utilized by a peptide decoder to generate amino acid sequences in an auto-regressive manner. Nonetheless, the mass spectra frequently lack informative peaks (McDonnell et al., 2022; Mao et al., 2023; Zhou et al., 2024; Yang et al., 2024), stemming from incomplete fragmentation of precursor peptides or inherent limitations within tandem mass spectrometer. This deficiency results in insufficient information for reconstructing peptide sequences. Consequently, existing methods struggle to effectively model the intricate spectral patterns, leading to potential gaps hindering the performance of *de novo* peptide sequencing, as illustrated in Figure 1.

To bridge this gap, we propose **L**atent **I**mputation before **P**rediction (**LIP**Novo), a new computational paradigm that revolutionizes the *de novo* sequencing pipeline by incorporating an imputation step. The imputation is formulated as a novel set prediction problem. Specifically, after modeling all pairwise interactions between observed peaks in a spectrum, LIPNovo integrates an imputation module, instantiated with a standard transformer decoder (Vaswani, 2017) and simple feed-forward networks, to predict latent representations of theoretical peaks corresponding to ideal fragmentation, specifically *b*- and *y*-ions of the target peptide (Zhou et al., 2024). By employing bipartite matching (Carion et al., 2020) to align predictions with ground truths and designing an imputation loss function to guide the imputation process, LIPNovo explicitly reconstructs missing fragmentation information beyond the observed incomplete spectrum. This capability stems from leveraging certain informative noise peaks (Tabb et al., 2003; Yang et al., 2024) and capturing the missing pattern from a large-scale training dataset. Through imputation, LIPNovo functions as a signal enhancement mechanism that directly reduces ambiguities, thus elucidating patterns and dependencies between the spectrum and the peptide, in stark contrast to previous methods that rely solely on incomplete spectra.

We build LIPNovo on top of the competitive CasaNovo baseline (Yilmaz et al., 2022; 2024). CasaNovo has recently undergone iterations with substantial performance improvements (Yilmaz et al., 2024). Experiments are conducted on three benchmark datasets (Zhou et al., 2024). The experimental results show that LIPNovo achieves significantly better results under different levels of missing fragmentation ratios, as demonstrated in Figure 1. Overall, LIPNovo surpasses CasaNovo by +5.6%, +20.0%, and +11.2% in amino acid precision on the Nine-species, Seven-species, and HC-PT datasets, respectively. Moreover, LIPNovo outperforms state-of-the-art methods by clear margins in amino acid-level, peptide-level, and post-translational modification (PTM)-level metrics. Additionally, we also test the model performance *w.r.t* the imputation quality, and the results suggest that the imputation quality correlates positively to the peptide sequencing performance. This further confirms the feasibility of our idea. To summarize, the core contributions of this work are as follows.

- We introduce LIPNovo, a new computational paradigm that incorporates a latent space imputation step into *de novo* peptide sequencing, effectively addressing the challenge of missing fragmentation information.

- LIPNovo formulates imputation as a novel set prediction problem, utilizing bipartite matching to align predicted latent representations with ground truths and a carefully designed imputation loss function to guide the imputation process.

- LIPNovo achieves substantial performance gains over existing state-of-the-art methods across multiple benchmark datasets, demonstrating its superiority.

## 2. Related Work

### 2.1. De Novo Peptide Sequencing

Deep learning techniques (LeCun et al., 2015) have significantly transformed the field of *de novo* peptide sequencing. Among them, DeepNovo (Tran et al., 2017) pioneered the use of a deep neural network that integrates CNN and LSTM networks. Then, PepNet (Liu et al., 2023) employed a fully CNN network, while GraphNovo (Mao et al., 2023) introduced a two-stage graph-based network that finds the optimal path among observed peaks to guide the sequence prediction. Recently, CasaNovo (Yilmaz et al., 2022) introduced powerful transformers for peptide sequencing, which greatly improves performance. Specifically, CasaNovo adopts an encode-then-decode pipeline that establishes direct mappings from observed spectra to peptides, now a mainstream framework for the task. Under this framework, Yilmaz et al. (Yilmaz et al., 2024) incorporated a beam search strategy (Freitag & Al-Onaizan, 2017) to improve

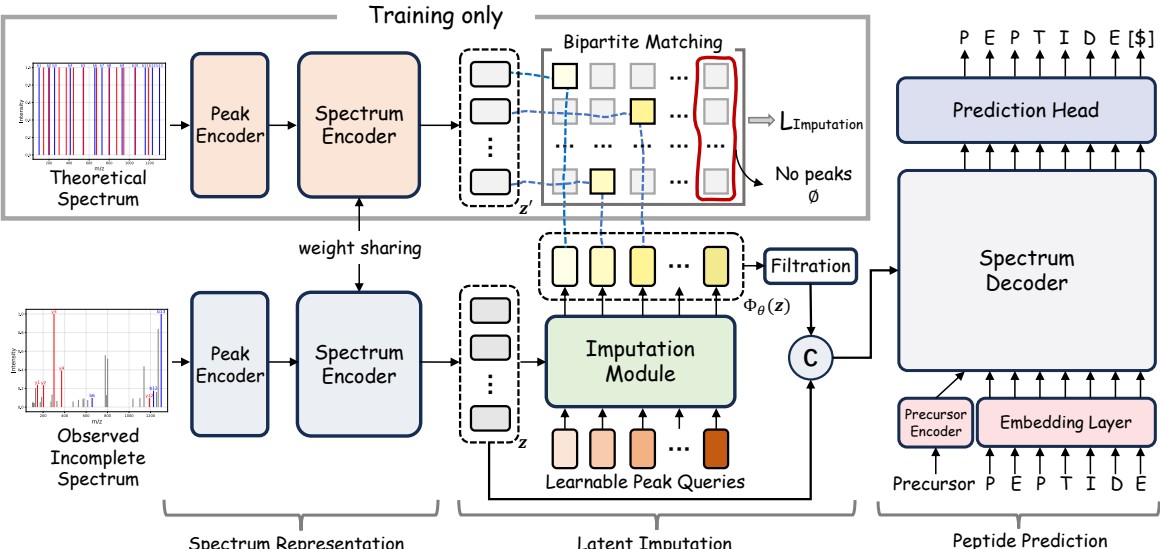

*Figure 2.* Illustration of the computational paradigm of LIPNovo. During training, LIPNovo generates a theoretical spectrum based on the target peptide (Figure 3), which is then embedded using the spectrum encoder, along with the observed spectrum. Then, LIPNovo learns to impute the latent representation of the theoretical peaks. Bipartite matching is utilized to enable unique matching between imputed results and ground truths, followed by a tailed imputation training objective $\mathcal{L}_{\text{Imputation}}$. Finally, the highly confident imputation results are concatenated with the original spectrum representations and input into the peptide decoder to predict the peptide sequence. During inference, the upper part is discarded, eliminating the need for the theoretical spectrum during testing. "[$]" is the stop token.

performance. Moreover, ContraNovo (Jin et al., 2024) incorporated contrastive learning and $\pi$-HelixNovo (Yang et al., 2024) generates complementary spectra as supplementary inputs, operating under the prior that *b*- and *y*-ions should appear symmetrically in the observed mass spectrum. In addition, AdaNovo (Xia et al., 2024) introduced an adaptive training algorithm that uses conditional mutual information to improve the identification of PTM. A comprehensive comparison of these methods is provided by NovoBench (Zhou et al., 2024) under fair experimental settings. Different from these methods, our approach directly imputes missing fragmentation information before prediction, forming a new end-to-end computational paradigm.

### 2.2. Missing Data Imputation

Missing data is a common challenge across various real-world applications, impacting the reliability of analyses. Imputation techniques (Sun et al., 2023; Li et al., 2024a) have proven essential in mitigating the issues associated with incomplete datasets. Early methods (Lin & Tsai, 2020) utilized traditional machine learning techniques for imputation, including K-Nearest Neighbors (KNN) (Zhang, 2012), MissForest imputation (Stekhoven & Bühlmann, 2012), and Multiple Imputation by Chained Equations (MICE) (White et al., 2011). Due to the ineffectiveness of these approaches in capturing inherent data patterns, recent studies (Sun et al., 2023) designed deep learning-based strategies to overcome these limitations. This includes utilizing generative adversarial networks (Goodfellow et al., 2020), variational auto-

encoders (Kingma, 2013), and diffusion models (Zheng & Charoenphakdee, 2022). Moreover, advanced methods that ensemble multiple imputation methods (Li et al., 2024b) or incorporate domain-specific knowledge (Yang et al., 2024; Hayat & Hasan, 2024) have been explored to enhance imputation accuracy and address various missing data mechanisms. In this work, we tackle the missing fragmentation issue in the *de novo* peptide sequencing task from an imputation perspective, based on the prior that certain noise ions can provide useful information for peptide recognition (Tabb et al., 2003; Yang et al., 2024) and that an explicit imputation objective can be obtained during training.

## 3. Methodology

### 3.1. Preliminary

*De novo* peptide sequencing is the process of translating the mass spectra obtained from the tandem mass spectrometer into amino acid sequences. As shown in Figure 2, the mass spectra are histograms showing the intensity plotted against the mass-to-charge ($m/z$) values of the ions, which result from the fragmentation of the intact peptides, usually known as the precursors.

Formally, a mass spectrum can be represented as $\boldsymbol{x} = \{(m_i\ I_i)\}_{i=1}^{N}$, where $(m_i, I_i)$ denotes the pair of $m/z$ and intensity, and $N$ signifies the number of peaks. The intensity is unitless but correlates monotonically with the quantity of ions contributing to the observed peaks (Yilmaz et al., 2022).

The $m/z$ values correspond to the prefixes (*i.e.*, $b$-ions) and suffixes (*i.e.*, $y$-ions) of the peptide sequence. Additionally, the precursor, denoted as $t = (m_{prec}, c_{prec})$ comprising the mass $m_{prec} \in \mathbb{R}$ and charge state $c_{prec} \in \{1, 2, ..., 10\}$, is also crucial for peptide identification, as the peptide mass should align within a specified tolerance of the total precursor mass. The peptide sequence can be denoted as $\boldsymbol{y} = \{y_l\}_{l=1}^{L}$, where each $y_l$ belongs to the amino acid vocabulary including 20 canonical amino acids and their post-translational modifications. $L$ is the peptide length that can vary among different peptides. Thus, *de novo* peptide sequencing requires a model that leverages $\boldsymbol{x}$ and $\boldsymbol{t}$ to predict the probability of $\boldsymbol{y}$. This can be formulated as the product of the conditional probabilities of each amino acid:

$$P(\boldsymbol{y} \mid \boldsymbol{x}, \boldsymbol{t}) = \prod_{l=1}^{L} P(y_l \mid \boldsymbol{y}_{<l}, \boldsymbol{x}, \boldsymbol{t}), \tag{1}$$

where $\boldsymbol{y}_{<l} = \{y_j\}_{j=1}^{l-1}$ denotes all amino acids that precede $y_l$ in the peptide sequence.

## 3.2. Latent Imputation before Prediction

In typical scenarios, the spectrum $\boldsymbol{x}$ may exhibit missing peaks that are crucial for peptide identification, resulting in performance degradation. To address this, the central idea of our work is to directly impute the missing information before peptide prediction. The overall pipeline of our LIP-Novo is shown in Figure 2, which comprises three steps: spectrum representation, latent space imputation, and the final peptide prediction.

### 3.2.1. SPECTRUM REPRESENTATION

Following CasaNovo (Yilmaz et al., 2022), we first transform $\boldsymbol{x} = \{(m_i, I_i)\}_{i=1}^{N}$ into a $d$-dimensional embedding using a peak encoder. Specifically, the peak encoder applies a fixed sinusoidal embedding (Vaswani, 2017) to each $m_i$, where each feature in the embedding $f \in \mathbb{R}^d$ is defined as

$$f_j = \begin{cases} \sin\left(m_i / \left(\frac{\lambda_{\max}}{\lambda_{\min}} \left(\frac{\lambda_{\min}}{2\pi}\right)^{2j/d}\right)\right), & \text{for } j \leq d/2 \\ \cos\left(m_i / \left(\frac{\lambda_{\max}}{\lambda_{\min}} \left(\frac{\lambda_{\min}}{2\pi}\right)^{2j/d}\right)\right), & \text{for } j > d/2 \end{cases}, \tag{2}$$

where, $\lambda_{\max}$ and $\lambda_{\min}$ are set to $10,000$ and $0.001$, respectively. Concurrently, $I_i$ is projected into the $d$-dimensional space via a linear layer. And the $m/z$ and intensity embeddings are integrated by summing them to generate the peak embedding.

Subsequently, a spectrum encoder, implemented using a standard transformer encoder, is employed to convert the peak embeddings into the latent representation space. The spectrum encoder utilizes the self-attention mechanism to capture and model the relationships between observed peaks.

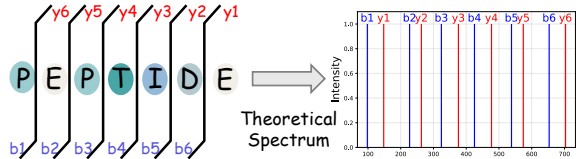

*Figure 3.* Illustration of theoretical spectrum calculation. For example, by splitting the position at 'E' and 'P', we can derive the b2 ion (PE) and the y5 ion (PTIDE). The masses of these two ions can be calculated using the mass table of amino acid residues. Here, we assume a charge of +1 and set the intensity to 100%.

As such, we can get a set of mass spectrum representation, denoted as $\boldsymbol{z} = \{z_i\}_{i=1}^{N}$, where each $z_i \in \mathbb{R}^d$. Notably, $\boldsymbol{z}$ is permutation invariant, meaning that the order of the peaks in the mass spectrum does not influence the peptide identification results.

### 3.2.2. LATENT SPACE IMPUTATION

After encoding the spectral features, traditional methods (Yilmaz et al., 2022; Xia et al., 2024; Yang et al., 2024; Yilmaz et al., 2024) directly predict the amino acid sequence from them. However, the observed spectrum usually exhibits varying degrees of missing informative peaks, making it challenging to rely solely on $\boldsymbol{z}$ for peptide prediction.

To address this issue, we introduce a latent space imputation step prior to peptide prediction. Specifically, we begin by calculating the mass value $m_j'$ of each ideal fragmentation (*i.e.*, one of all $b$- and $y$-ions) derived from splitting the precursor peptide, as shown in Figure 3. Then, the intensity value is estimated as $I_j' = \max\{I_1, I_2, ..., I_N\}$, based on the prior that fragmentations should occur in consistently high quantities in an ideal scenario (Yang et al., 2024). Therefore, the theoretical spectrum can be represented as $\boldsymbol{x}' = \{m_j', I_j'\}_{j=1}^{N'}$, where the charge state is assumed to be +1, and $N' = 2(L-1)$ denotes the number of ideal fragmentation.

Afterwards, we encode $\boldsymbol{x}'$ using the same spectrum encoding method discussed above, generating the theoretical spectrum representation $\boldsymbol{z}' = \{z_j'\}_{j=1}^{N'}$. $\boldsymbol{z}'$ contains sufficient information to facilitate peptide prediction, and can therefore serve as the imputation objective for complementing $\boldsymbol{z}$. Thus, we construct an imputation module $\Phi_\theta(\cdot)$ parameterized by $\theta$, which takes $\boldsymbol{z}$ as input and directly produces a set of predictions, yielding the following optimization objective:

$$\hat{\theta} = \arg\min_{\theta} \mathcal{L}_{\text{Imputation}}\left(\Phi_\theta(\boldsymbol{z}), \boldsymbol{z}'\right), \tag{3}$$

where $\mathcal{L}_{\text{Imputation}}$ represents a suitable distance metric.

### 3.2.3. IMPUTATION VIA BIPARTITE MATCHING

In Eq. (3), the target $z'$ typically varies in length across different peptide instances, presenting a variable-length set prediction challenge. To tackle this, our LIPNovo generates a fixed-size set of $M$ predictions in a single forward pass through the imputation module, where $M$ is a hyperparameter chosen to exceed the typical number of theoretical peaks in a spectrum. A key challenge during training is evaluating $\Phi_\theta(z)$ with respect to $z'$ to ensure a unique matching between the two sets. To overcome this, we employ the optimal *bipartite matching* between predictions and ground truths, followed by optimizing peak-specific losses.

Specifically, given $\Phi_\theta(z) \in \mathbb{R}^{M \times d}$ and $z' \in \mathbb{R}^{N' \times d}$ with $M > N'$, we first pad $z'$ by appending $\varnothing$ to the end of it to align its length with $\Phi_\theta(z)$. To find a bipartite matching between $\Phi_\theta(z)$ and $z'$, inspired by (Carion et al., 2020), we search for a permutation of $M$ elements $\sigma \in \boldsymbol{\sigma}_M$ with the lowest cost:

$$\hat{\sigma} = \arg \min_{\sigma \in \boldsymbol{\sigma}_M} \sum_{j=1}^{M} \mathcal{L}_{\text{Match}} \left( \Phi_\theta(z)_{\sigma(j)}, z'_j \right), \qquad (4)$$

where $\sigma(j)$ indexes a prediction from the imputation module, and $\mathcal{L}_{\text{Match}}$ measures the matching cost between the ground truth and the indexed prediction. In calculating $\mathcal{L}_{\text{Match}}$, we consider the pairwise mean squared error (MSE) between predictions and targets, as well as the probability of whether each prediction is responsible for a true target or not (*i.e.*, $\varnothing$). Thus, denoting $p_{\sigma(j)} \in [0, 1]$ as the probability associated with a prediction indexed by $\sigma(j)$, we define the matching cost as

$$\mathcal{L}_{\text{Match}}(\Phi_\theta(z)_{\sigma(j)}, z'_j) = \|\Phi_\theta(z)_{\sigma(j)} - z'_j\|^2 + $$
$$1 - (\mathbb{1}_{z'_j \notin \varnothing} p_{\sigma(j)} + \mathbb{1}_{z'_j \in \varnothing}(1 - p_{\sigma(j)})), \tag{5}$$

where $\mathbb{1}$ represents the indicator function.

In practice, the optimal assignment can be efficiently computed by the Hungarian algorithm (Kuhn, 1955). After solving $\hat{\sigma}$, we calculate the imputation loss in Eq. (3) as

$$\mathcal{L}_{\text{Imputation}} \left( \Phi_\theta(z), z' \right) = \frac{1}{N'} \sum_{j=1}^{N'} \|\Phi_\theta(z)_{\hat{\sigma}(j)} - z'_j\|^2 + $$
$$\frac{1}{M} \left[ -\sum_{j=1}^{N'} \log p_{\hat{\sigma}(j)} - \sum_{j=N'+1}^{M} \log \left( 1 - p_{\hat{\sigma}(j)} \right) \right], \tag{6}$$

where the first term computes the MSE between matched pairs, while the second term aims to maximize the probability of predictions corresponding to the ground truth and minimize the probability of those corresponding to $\varnothing$.

### 3.2.4. ARCHITECTURE OF IMPUTATION MODULE

We then illustrate the architecture of $\Phi_\theta$. As mentioned previously, two types of outputs (*i.e.* the imputed results and their probabilities) are required by the imputation process. To this end, we implement $\Phi_\theta$ using a two-branch architecture that shares a common imputation decoder. The imputation decoder is built using a standard transformer decoder. To accommodate set prediction, we introduce a set of *learnable* query vectors, denoted as $\boldsymbol{q} = \{q_j\}_{j=1}^M$, where each $q_j \in \mathbb{R}^d$. These query vectors are randomly initialized and input to the imputation decoder, with the observed spectrum representations $z$ as encoded memories to produce a total of $M$ embeddings, as illustrated in Figure 2. On top of them, two separate feed-forward networks (FFN) are applied: one generates the imputed representations, while the other (appended with sigmoid activation) produces the probabilities required in Eq. (6). This design is simple and efficient, enabling the parallelized prediction of latent representations for all theoretical peaks, thus providing an effective instantiation for $\Phi_\theta$.

### 3.2.5. PEPTIDE PREDICTION

After the imputation step, we combine $z$ with $\Phi_\theta(z)$ to predict the amino acid sequence. Note that the set $\Phi_\theta(z)$ may contain predictions associated with $\varnothing$. To address this, we apply a probability threshold $\tau$ to filter $\Phi_\theta(z)$, resulting in a refined set of predictions:

$$\tilde{z} = \{z_j \mid z_j \in \Phi_\theta(z) \wedge p_j > \tau\}, \tag{7}$$

where $p_j$ denotes the predicted probability for $z_j$. $\tilde{z}$ is then combined with $z$ and fed into a peptide decoder to predict the amino acid sequence in an auto-regressive manner. The peptide decoder is also a transformer decoder, followed by a prediction head composed of a linear layer and a softmax activation function. As in previous works (Yilmaz et al., 2022), the serial prediction process begins with the precursor $t$ as input, which is embedded using the same method as the spectrum. The prediction is supervised by the amino acid cross-entropy (CE) loss $\mathcal{L}_{\text{CE}}([\tilde{z}; z])$, where $\mathcal{L}_{\text{CE}}$ is defined as:

$$\mathcal{L}_{\text{CE}}(\boldsymbol{c}) = -\sum_{j=1}^{L} \log P \left( y_j \mid \boldsymbol{y}_{<j}, \boldsymbol{c}, \boldsymbol{t} \right). \tag{8}$$

Furthermore, to generate meaningful theoretical spectrum representations $z'$ as imputation targets, we utilize the same peptide decoder to decode $z'$ and employ the CE loss (*i.e.*, $\mathcal{L}_{\text{CE}}(z')$) for supervision. Finally, the total training objective of LIPNovo is the combination of three losses:

$$\mathcal{L}_{\text{total}} = \mathcal{L}_{\text{CE}}([\tilde{z}; z]) + \mathcal{L}_{\text{CE}}(z') + \mathcal{L}_{\text{Imputation}}. \tag{9}$$

*Remark* 3.1. After training, LIPNovo is capable of reconstructing missing information from incomplete spectra during testing. This capability stems from two perspectives.

*Table 1.* Empirical comparison with state-of-the-art methods on Nine-species, Seven-species, and HC-PT datasets in amino acid-level and peptide-level performance. † denotes our retrained results, and other results are provided by NovoBench. The best is marked in bold.

| Method | Amino Acid-Level Performance | | | | | | Peptide-Level Performance | | | | | |
| | Nine-species | | Seven-species | | HC-PT | | Nine-species | | Seven-species | | HC-PT | |
| | Prec. | Recall | Prec. | Recall | Prec. | Recall | Prec. | AUC | Prec. | AUC | Prec. | AUC |
|---|---|---|---|---|---|---|---|---|---|---|---|---|
| PEAKS (Ma et al., 2003) | 0.748 | - | - | - | - | - | 0.428 | - | - | - | - | - |
| DeepNovo (Tran et al., 2017) | 0.696 | 0.638 | 0.492 | 0.433 | 0.531 | 0.534 | 0.428 | 0.376 | 0.204 | 0.136 | 0.313 | 0.255 |
| PointNovo (Qiao et al., 2021) | 0.740 | 0.671 | 0.196 | 0.169 | 0.623 | 0.622 | 0.480 | 0.436 | 0.022 | 0.007 | 0.419 | 0.373 |
| InstaNovo (Eloff et al., 2023) | 0.420 | 0.395 | 0.192 | 0.176 | 0.289 | 0.285 | 0.164 | 0.123 | 0.031 | 0.009 | 0.057 | 0.034 |
| CasaNovo (Yilmaz et al., 2024) | 0.697 | 0.696 | 0.322 | 0.327 | 0.442 | 0.453 | 0.481 | 0.439 | 0.119 | 0.084 | 0.211 | 0.177 |
| AdaNovo (Xia et al., 2024) | 0.698 | 0.709 | 0.379 | 0.385 | 0.442 | 0.451 | 0.505 | 0.469 | 0.174 | 0.135 | 0.212 | 0.178 |
| AdaNovo† (Xia et al., 2024) | 0.681 | 0.681 | 0.403 | 0.405 | 0.492 | 0.496 | 0.473 | 0.439 | 0.189 | 0.149 | 0.289 | 0.254 |
| $\pi$-HelixNovo (Yang et al., 2024) | 0.765 | 0.758 | 0.481 | 0.472 | 0.588 | 0.582 | 0.517 | 0.453 | 0.234 | 0.173 | 0.356 | 0.318 |
| $\pi$-HelixNovo† (Yang et al., 2024) | 0.765 | 0.752 | 0.465 | 0.462 | 0.532 | 0.537 | 0.509 | 0.431 | 0.218 | 0.156 | 0.301 | 0.261 |
| Baseline† (Yilmaz et al., 2024) | 0.741 | 0.740 | 0.357 | 0.366 | 0.525 | 0.530 | 0.529 | 0.493 | 0.159 | 0.119 | 0.324 | 0.290 |
| **LIPNovo (Ours)** | **0.797** | **0.797** | **0.557** | **0.560** | **0.637** | **0.643** | **0.582** | **0.547** | **0.327** | **0.281** | **0.458** | **0.427** |

*Table 2.* Empirical comparison with state-of-the-art methods on Nine-species, Seven-species, and HC-PT datasets in PTM-level performance. † denotes our retrained results, and other results are given by NovoBench. The best results are marked in bold.

| Method | Nine-species | | Seven-species | | HC-PT | |
| | Prec. | Recall | Prec. | Recall | Prec. | Recall |
|---|---|---|---|---|---|---|
| DeepNovo | 0.576 | 0.529 | 0.391 | 0.373 | 0.626 | 0.615 |
| PointNovo | 0.629 | 0.546 | 0.117 | 0.094 | 0.676 | 0.740 |
| InstaNovo | 0.443 | 0.294 | 0.126 | 0.115 | 0.350 | 0.261 |
| CasaNovo | 0.706 | 0.566 | 0.360 | 0.251 | 0.501 | 0.460 |
| AdaNovo | 0.652 | 0.570 | 0.448 | 0.321 | 0.552 | 0.482 |
| AdaNovo† | 0.678 | 0.552 | 0.430 | 0.356 | 0.562 | 0.532 |
| $\pi$-HelixNovo | 0.680 | 0.598 | 0.473 | 0.366 | 0.568 | 0.667 |
| $\pi$-HelixNovo† | 0.723 | 0.593 | 0.362 | 0.370 | 0.632 | 0.566 |
| Baseline† | 0.755 | 0.601 | 0.368 | 0.292 | 0.550 | 0.582 |
| **LIPNovo (Ours)** | **0.765** | **0.656** | **0.604** | **0.498** | **0.732** | **0.745** |

From a local perspective, some noise peaks in the spectrum correspond to informative ions (Yang et al., 2024; Tabb et al., 2003), which arise due to incorrect cleavage patterns of theoretical ions. Therefore, these peaks can still provide valuable information for imputation. From a global perspective, although a single spectrum instance may lack sufficient information to reconstruct missing peaks, the imputation model is trained across the entire dataset. This allows the model to learn patterns of missing information by leveraging cross-instance correlations, enabling it to infer the latent representations of missing fragments. The imputation module explicitly reconstructs the missing fragments in the latent space, which reduces ambiguity in the incomplete spectra, and enables the model to better capture the dependencies between the spectrum and the peptide sequence.

## 4. Experiment

### 4.1. Experimental Setup

**Datasets.** Following the benchmark (Zhou et al., 2024), we conduct experiments on three datasets: Nine-species (Tran et al., 2017), Seven-species (Tran et al., 2017), and HC-PT (Eloff et al., 2023). The Nine-species dataset, the most widely used in previous studies, contains mass spectra from nine different species. The yeast species is utilized as the test set, while the other eight species are used for training and evaluation. The Seven-species dataset consists of mass spectra from seven species, where the yeast species is set as the test set and the other six species are used for training and evaluation. The HC-PT dataset contains mass spectra of human-origin peptides, including synthetic tryptic peptides that represent all canonical human proteins and isoforms, as well as peptides generated by alternative proteases and human leukocyte antigen peptides.

**Evaluation Metrics.** We evaluate the peptide sequencing performance using amino acid-level precision and recall, peptide-level precision and area under the precision-recall curve (AUC), as well as precision and recall of PTM identification.

### 4.2. Implementation Details

Both the spectrum encoder and peptide decoder consist of 9 layers, with a hidden dimension of 512, an FFN layer dimension of 1024, and 8 attention heads. The maximum number of peaks is set to 150, while the maximum peptide length is 100. Following (Yang et al., 2024), we employ the complementary spectrum as additional input. The amino acid vocabulary comprises 20 canonical amino acids and 3 PTMs (oxidation of methionine and deamidation of asparagine or glutamine), in addition to a stop token [$] used to indicate the end of the sequence. For the imputation module, the imputation decoder has 3 layers, and the number of peak queries is set to 100. The filter threshold $\tau$ is set to 0.8. Models are trained with a batch size of 32 for 30 epochs. The learning rate is 5e-4, with a weight decay of 1e-5. The learning rate is linearly increased from zero to the peak value in 100k warm-up steps, followed by a cosine-shaped decay. For $\mathcal{L}_{CE}$, we use label smoothing of 0.01. During inference, a beam search strategy with a beam size of 5 is utilized for all models. The parameters are kept consistent

*Table 3.* Leave-one-out cross validation compared to baseline on the Nine-species dataset. † means our re-trained results.

| Species | Method | Amino Acid | | Peptide | | PTM | |
|---|---|---|---|---|---|---|---|
| | | Prec. | Recall | Prec. | AUC | Prec. | Recall |
| Bacillus | Baseline[†] | 0.743 | 0.745 | 0.559 | 0.523 | 0.797 | 0.689 |
| | **LIPNovo** | 0.806 | 0.807 | 0.607 | 0.581 | 0.855 | 0.739 |
| Clam Ba. | Baseline[†] | 0.754 | 0.752 | 0.544 | 0.509 | 0.742 | 0.591 |
| | **LIPNovo** | 0.805 | 0.805 | 0.591 | 0.563 | 0.764 | 0.647 |
| Honeyb. | Baseline[†] | 0.761 | 0.757 | 0.554 | 0.519 | 0.762 | 0.624 |
| | **LIPNovo** | 0.807 | 0.806 | 0.606 | 0.577 | 0.769 | 0.675 |
| Human | Baseline[†] | 0.769 | 0.770 | 0.575 | 0.540 | 0.784 | 0.612 |
| | **LIPNovo** | 0.805 | 0.805 | 0.596 | 0.567 | 0.772 | 0.672 |
| M.mazei | Baseline[†] | 0.754 | 0.754 | 0.558 | 0.532 | 0.779 | 0.582 |
| | **LIPNovo** | 0.807 | 0.807 | 0.596 | 0.565 | 0.812 | 0.630 |
| Mouse | Baseline[†] | 0.753 | 0.752 | 0.558 | 0.521 | 0.799 | 0.608 |
| | **LIPNovo** | 0.808 | 0.807 | 0.607 | 0.579 | 0.803 | 0.667 |
| Ricebean | Baseline[†] | 0.753 | 0.755 | 0.549 | 0.510 | 0.760 | 0.593 |
| | **LIPNovo** | 0.794 | 0.796 | 0.577 | 0.545 | 0.764 | 0.634 |
| Tomato | Baseline[†] | 0.717 | 0.716 | 0.506 | 0.464 | 0.710 | 0.540 |
| | **LIPNovo** | 0.800 | 0.798 | 0.577 | 0.544 | 0.777 | 0.627 |
| **Mean** | Baseline[†] | 0.751 | 0.750 | 0.550 | 0.515 | 0.767 | 0.605 |
| | **LIPNovo** | **0.804** | **0.804** | **0.595** | **0.565** | **0.790** | **0.661** |

*Table 4.* Comparison with GraphNovo (Mao et al., 2023).

| Method | # Params | Peptide Level | | Amino Acid Level | |
|---|---|---|---|---|---|
| | | Recall | AUC | Recall | Prec. |
| GraphNovo | 88.2M | 0.712 | - | 0.876 | 0.874 |
| LIPNovo | 68.4M | **0.729** | **0.707** | **0.876** | **0.882** |

for all datasets. All experiments were performed using an NVIDIA GeForce RTX 4090 GPU.

### 4.3. Comparison with State-of-the-arts

We compare LIPNovo with several established *de novo* sequencing competitors, including DeepNovo (Tran et al., 2017), PointNovo (Qiao et al., 2021), InstaNovo (Eloff et al., 2023), CasaNovo (Yilmaz et al., 2024), AdaNovo (Xia et al., 2024), and $\pi$-HelixNovo (Yang et al., 2024). Notably, we retrain a CasaNovo as the direct baseline of our LIPNovo with the same configurations. The results are summarized in Tables 1 and 2.

**Amino Acid-Level Comparison.** LIPNovo achieves the highest amino acid-level precision and recall across all datasets. Specifically, on the Nine-species dataset, LIPNovo reaches a precision of 0.797 and a recall of 0.797, outperforming the closest competitor $\pi$-HelixNovo by +3.2% and +4.5%. It also surpasses the powerful software PEAKS by +5.1% in precision. In the Seven-species and HC-PT, LIPNovo also sets a new record with a high performance.

**Peptide-Level Comparison.** Peptide-level performance are crucial for assessing the practical utility of the models, as the primary objective of the peptide sequencing task is to accurately assign a complete peptide sequence to each observed spectrum. As shown in Table 1, LIPNovo substantially outperforms previous methods across all datasets in the peptide level, achieving mean improvements of +13.9%

*Table 5.* Component ablation. "Impu." denotes the imputation module, and $\mathcal{L}_{CE}(z')$ means the CE loss supervised on the theoretical spectrum. "Comp." means the complementary spectrum.

| | Baseline | Impu. | $\mathcal{L}_{CE}(z')$ | Comp. | Amino Acid Level | | Peptide Level | |
|---|---|---|---|---|---|---|---|---|
| | | | | | Prec. | Recall | Prec. | AUC |
| 1 | ✓ | ✗ | ✗ | ✗ | 0.741 | 0.740 | 0.529 | 0.493 |
| 2 | ✓ | ✗ | ✗ | ✓ | 0.755 | 0.755 | 0.537 | 0.500 |
| 3 | ✓ | ✓ | ✗ | ✓ | 0.766 | 0.764 | 0.546 | 0.513 |
| 4 | ✓ | ✓ | ✓ | ✗ | 0.782 | 0.782 | 0.569 | 0.536 |
| 5 | ✓ | ✓ | ✓ | ✓ | **0.797** | **0.797** | **0.582** | **0.547** |

in precision compared to AdaNovo and +11.3% compared to $\pi$-HelixNovo. Additionally, LIPNovo demonstrates the highest AUC among the compared methods. It also exceeds PointNovo, the second-best method on the HC-PT dataset, by +5.4%. These results highlight the superiority of LIPNovo as a powerful *de novo* sequencing method, reinforcing its potential as a new computational paradigm in training peptide sequencing models.

**PTM-Level Comparison.** PTMs represent a critical aspect of protein function and regulation, making PTM-level comparison valuable for evaluating the efficacy of sequencing models (Xia et al., 2024). In this regard, LIPNovo also exhibits remarkable performance. Specifically, as shown in Table 2, LIPNovo outperforms AdaNovo by +8.7%, +17.4% and +17.0% in PTM precision across three datasets. Additionally, benefiting from the imputation mechanism, LIPNovo achieves significant improvements in PTM recall, with increases of +5.5% and +12.5% compared to the second-best method on Nine- and Seven-species datasets, respectively.

**Leave-One-Out Cross Validation.** We also perform leave-one-out cross-validation that takes turns selecting a species as the test set and trains on the other eight species. The results are shown in Table 3, where "mean" is computed by averaging across the eight test species. LIPNovo consistently demonstrates improvements in most scenarios. On average, LIPNovo outperforms the baseline by +5.3%, +4.5%, and +2.3% in precision across the three performance levels.

**Comparison with GraphNovo.** We also compare LIPNovo with GraphNovo (Mao et al., 2023). Training GraphNovo is resource-intensive, making it impractical to retrain on benchmark datasets. To ensure a fair comparison, we trained LIPNovo on the dataset collected in GraphNovo. As shown in Table 4, LIPNovo outperforms GraphNovo by +1.7% in peptide recall. Notably, GraphNovo is a two-stage model that is not only computationally demanding but also has a larger parameter count. In contrast, LIPNovo offers a simpler and more efficient end-to-end solution while maintaining superior performance, demonstrating its effectiveness and practicality.

### 4.4. Ablation Study

**Component Ablation.** We evaluate the contribution of each component in LIPNovo on the Nine-species dataset. The

*Table 6.* Sensitivity to hyper-parameters. "Prec.A" and "Prec.P" means amino acid and peptide precision, respectively.

| $M$ | Prec.A | Prec.P | **Layer** | Prec.A | Prec.P | $\tau$ | Prec.A | Prec.P |
|-----|--------|--------|-----------|--------|--------|--------|--------|--------|
| 0 | 0.741 | 0.529 | 0 | 0.741 | 0.529 | 0.6 | 0.790 | 0.571 |
| 50 | 0.787 | 0.564 | 1 | 0.796 | 0.574 | 0.7 | 0.794 | 0.569 |
| **100** | 0.797 | 0.582 | **3** | 0.797 | 0.582 | **0.8** | 0.797 | 0.582 |
| 150 | 0.792 | 0.574 | 6 | 0.793 | 0.579 | 0.9 | 0.785 | 0.569 |

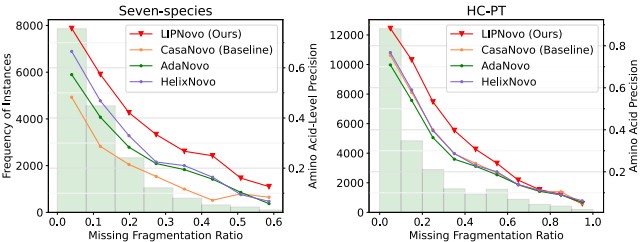

*Figure 4.* Missing fragmentation ratio *vs.* model performance. LIPNovo outperforms existing methods under various missing fragmentation ratios on Seven-species and HC-PT datasets.

results are presented in Table 5. Comparing rows 4 and 1, we observe that using the imputation mechanism alone leads to significant improvements. The $L_{CE}(z')$, which supervises the learning of theoretical spectrum representations, is also crucial; its removal results in a noticeable performance drop (row 4 vs. row 3). We then test the impact of complementary spectrum (Yang et al., 2024). Comparing row 5 and row 4, we find that it contributes an additional +1.5% increase in amino acid precision and +1.3% in peptide AUC. Overall, these results demonstrate the performance of each module, with the primary performance improvement stemming from our proposed imputation mechanism.

**Hyper-Parameter Sensitivity Analysis.** We then test the sensitivity of LIPNovo to three key hyper-parameters: the number of peak queries $M$, the layer count of the imputation module, and the threshold $\tau$ in Eq. (7). The experiments, conducted on the Nine-species dataset, are detailed in Table 6. For $M$, if the number of theoretical peaks exceeds $M$, we only choose the fist $M$ peaks as imputation targets. The results indicate that $M$=100 yields the optimal performance, with no further performance gains observed with higher values. Additionally, we find that setting the layer count to 3 and $\tau$ to 0.8 results in the best performance. These parameters are maintained consistently on other datasets.

### 4.5. Analysis

**LIPNovo effectively mitigates the impact of missing fragments.** We further investigate the performance in handling various missing fragmentation ratios, which is calculated by dividing the number of missing peaks by the number of ideal peaks in a spectrum, as in (Zhou et al., 2024). As shown in Figure 1 and 4, LIPNovo consistently achieves performance improvements under almost all missing ratios. An intriguing observation is that as the missing ratio reaches a very

*Table 7.* Parameters *vs.* model performance. ¶ is the extension.

| Method | # Params | Amino Acid Level | | Peptide Level | |
|--------|----------|-------|--------|-------|-------|
| | | Prec. | Recall | Prec. | AUC |
| Baseline | 47.4M | 0.741 | 0.740 | 0.529 | 0.493 |
| Baseline¶ | 69.4M | 0.750 | 0.751 | 0.539 | 0.494 |
| LIPNovo | 68.4M | **0.797** | **0.797** | **0.582** | **0.547** |

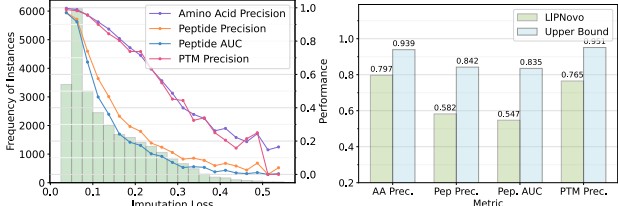

*Figure 5.* Imputation quality *vs.* model performance. (Left) A smaller imputation loss corresponds to higher performance. (Right) The upper bound of LIPNovo, obtained by directly using ground truth representations instead of predicted representations for the theoretical spectrum on the test set.

high value, the degree of improvement becomes limited. For instance, on the HC-PT dataset, at missing ratios ranging from 0.7 to 0.8, the improvement is merely +0.8%. We attribute this to the significant loss of signal peaks, which hinders effective imputation for the representations of theoretical peaks, consequently leading to reduced accuracy. Addressing this challenge represents potential future work, focusing on compensating for missing information under conditions of exceptionally high fragmentation proportions. This includes leveraging additional information, such as cross-sample correlations, and designing more effective imputation mechanisms.

**Enhancing imputation quality can improve peptide sequencing performance.** We visualize the imputation loss values *vs.* sequencing performance in Figure 5. The results are based on the test set of the Nine-species dataset. Note that the model predictions are not directly influenced by the imputation loss, as the imputation loss exclusively guides the generation of theoretical spectrum representations rather than peptide prediction. The figure shows that when the imputation loss is minimal (indicating high imputation quality), the model achieves exceptional performance. Conversely, as the imputation quality deteriorates, the model performance also declines. This suggests that enhancing the accuracy of imputation can aid in improving sequencing performance. In Figure 5, we also show the upper bound performance of LIPNovo, which is obtained by directly using ground truth representations of theoretical spectra instead of imputed results for peptide prediction during testing. The upper bound achieves an amino acid precision of 0.939 and a peptide precision of 0.842. This further validates the feasibility of our core idea that improving imputation quality is an effective way to boost peptide sequencing.

**The performance enhancement originates from the imputation mechanism rather than the additional parame-**

**ters.** While LIPNovo integrates an imputation module that introduces extra training parameters, this raises the question of whether the performance boost is merely due to the increased parameter number. To address this, we extend the baseline model by increasing the number of layers to 12 and adding an FFN to roughly match LIPNovo's parameter count. As shown in Table 7, this extension results in only a marginal improvement, whereas LIPNovo still outperforms it by a significant margin.

## 5. Conclusion

In this work, we present LIPNovo, a new computational framework that enhances *de novo* peptide sequencing through imputation. We frame imputation as a novel set prediction problem, utilizing optimal bipartite matching and a tailed imputation loss function. Experiments on three benchmark datasets demonstrate that LIPNovo achieves state-of-the-art performance across a wide range of metrics, highlighting its potential to advance proteomics research.

## Acknowledgements

This work was partially supported by RGC Collaborative Research Fund (No. C5055-24G), the Start-up Fund of The Hong Kong Polytechnic University (No. P0045999), the Seed Fund of the Research Institute for Smart Ageing (No. P0050946), and Tsinghua-PolyU Joint Research Initiative Fund (No. P0056509).

## Impact Statement

This paper presents work whose goal is to advance the field of Machine Learning. There are many potential societal consequences of our work, none of which we feel must be specifically highlighted here.

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

# Appendix

## A. Background

To assist readers who may not be familiar with proteomics, particularly tandem mass spectrometry analysis and the task of de novo peptide sequencing, we provide a brief background overview. Proteomics is the large-scale study of the structure, function, and interactions of proteins within a biological system, aimed at understanding cellular processes and biological mechanisms (Blackstock & Weir, 1999). An essential component of proteomics is the identification of proteins in biological samples (*e.g.*, within a living organism). Tandem mass spectrometry has become the primary high-throughput technique for protein identification. As illustrated in Figure 6, the standard shotgun proteomics workflow (Wolters et al., 2001) starts with enzymatic digestion of proteins, generating a mixture of peptides. These peptides, known as precursors, are then analyzed by a mass spectrometer, which measures their mass-to-charge ($m/z$) ratios during the first scan (MS1). The peptides are subsequently fragmented using techniques such as collision-induced dissociation (CID)(Yates et al., 1995) and higher-energy collisional dissociation (HCD)(Olsen & Mann, 2004), producing fragments that are analyzed in a second scan (MS2). During this fragmentation, peptides break randomly along their backbone, creating ions that correspond to the peptide's prefixes (*i.e.*, $b$-ions) and suffixes (*i.e.*, $y$-ions). The resulting MS2 spectrum consists of peaks characterized by $m/z$ values and intensities. While $m/z$ values are measured with high precision, intensity values, although less precise, are proportional to the number of ions contributing to each peak. The task of *de novo* peptide sequencing involves developing machine learning models that take MS2 data, along with corresponding precursor information, as input to predict the peptide sequence responsible for generating the observed mass spectrum. This prediction is followed by the assembly of various peptide segments to determine the complete protein sequence. The whole process is critical for decoding the complexities of the proteome, offering deep insights into the molecular underpinnings of biological systems and advancing our understanding of cellular functions and disease mechanisms.

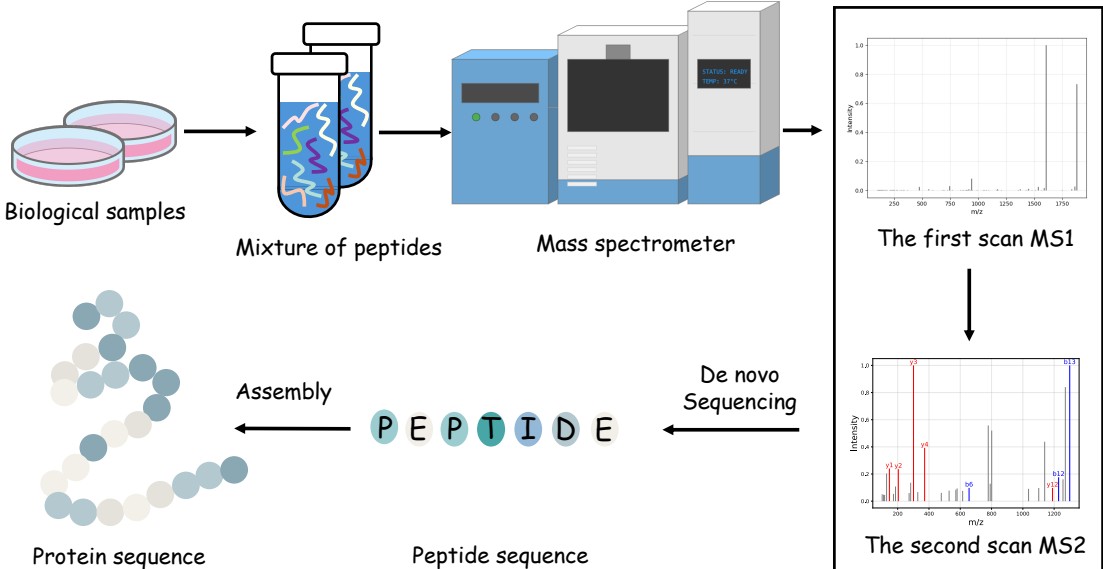

*Figure 6.* Illustration of the identification workflow of shotgun proteomics (Wolters et al., 2001).

## B. More Details

Table 8 presents comprehensive information on the dataset partitioning to assist in replicating our work. The average peptide length is obtained from NovoBench (Zhou et al., 2024).

## C. More Results

Figure 7 provides a detailed comparison of LIPNovo with other methods across three datasets, analyzing amino acid-level recall and peptide-level recall under varying missing fragmentation ratios. The results demonstrate that LIPNovo consistently outperforms existing approaches.

*Table 8.* Statistics of three benchmark datasets (Zhou et al., 2024).

| Dataset | # Training | # Validation | # Testing | Average Peptide Length | PTM class |
|---|---|---|---|---|---|
| Nine-species (Tran et al., 2017) | 499, 402 | 28, 572 | 27, 142 | 15.01 | 3 |
| Seven-species (Tran et al., 2017) | 317, 009 | 17, 740 | 17, 094 | 15.79 | 3 |
| HC-PT (Eloff et al., 2023) | 213, 284 | 25, 718 | 26, 536 | 12.53 | 1 |

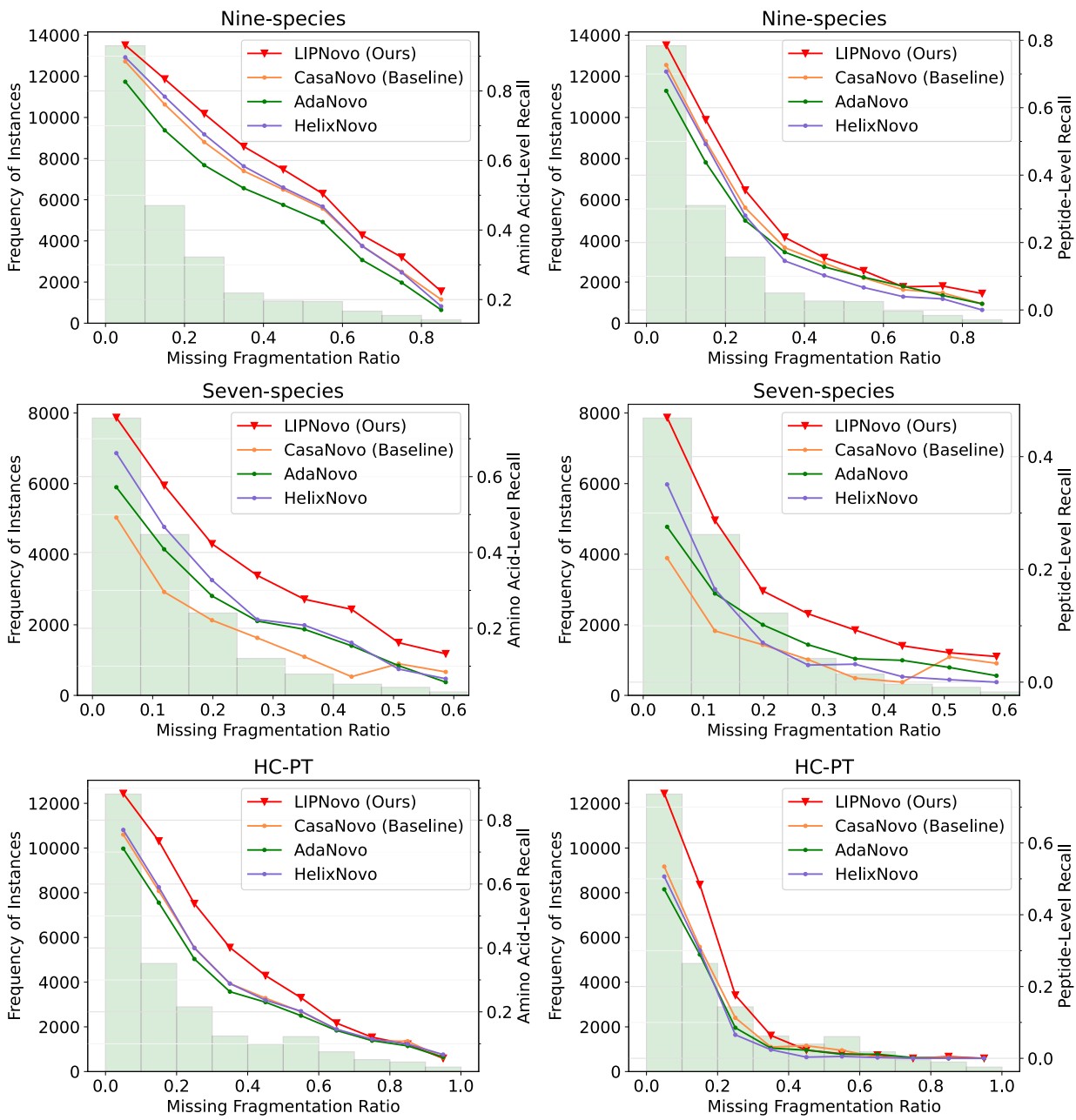

*Figure 7.* Comparison of amino acid recall (left column) and peptide recall (right column) under different missing fragmentation ratios between LIPNovo and state-of-the-art methods on three datasets.

