# OpenReview forum: "Latent Imputation before Prediction: A New Computational Paradigm for De Novo Peptide Sequencing"
_ICML.cc/2025/Conference — ICML 2025 poster_

### Official Review · Reviewer_2qTi · 2025-03-07

**Overall Recommendation:** 3

**Summary:**

This work aims to design peptide sequence based on observed mass spectra, addressing the issue of missing fragmentation. This problem is due to the incomplete fragmentation of precursor peptides or inherent limitations within tandem mass spectrometer. The author design a bipartite matching algorithm to impute the missing information.

**Claims And Evidence:**

Yes.

**Essential References Not Discussed:**

None.

**Experimental Designs Or Analyses:**

The experimental results show great performance improvement. The authors design extensive experiments and ablation studies to show the validness.

**Methods And Evaluation Criteria:**

The proposed algorithm, Latent Imputation before Prediction, generally makes sense and is well-motivated.

**Other Comments Or Suggestions:**

None.

**Other Strengths And Weaknesses:**

1. The motivation is quite clear and supported by the experimental results.

2. The design of proposed algorithms is generally reasonable.

3. The weakness of this work may can be the overall algorithm design is not that impressive and cannot be adapted to other fields. However, personally I like such application paper with simple yet effective approach.

4. The imputation may also introduce additional time consumption. Could the authors also provide the time portion of imputation in the entire pipeline.

**Questions For Authors:**

None.

**Relation To Broader Scientific Literature:**

None.

**Theoretical Claims:**

This work does not have theoretical claim.

---

> ### Author Rebuttal · Authors · 2025-03-30
>
> We acknowledge with gratitude for the reviewer's appreciation that our work's motivation is quite clear and supported by the experimental results, as well as the recognition that the design of proposed algorithm is generally reasonable. We address the reviewer's concerns in detail below.
>
> **Q1: The weakness of this work may can be the overall algorithm design is not that impressive and cannot be adapted
> to other fields. However, personally I like such application paper with simple yet effective approach.**
>
> Thanks for the constructive feedback and appreciation of our simple yet effective approach. While our algorithm is specifically  designed for peptide sequencing, it may offer insights into other computational biology areas facing similar challenges.  For instance, in single-cell transcriptomics [3], expression values appear undetected due to technical limitations rather than true biological absence, and the number of missing expression values varies across cells. Our method could serve as inspiration for developing analogous solutions in such contexts.
>
> **Q2: The imputation may also introduce additional time consumption. Could the authors also provide the time portion of imputation in the entire pipeline.**
>
> Thanks for the valuable suggestion. We evaluate the computational overhead of the imputation module in **Table 4**, which reports the **inference time and proportion** (averaged over the Nine-Species test set) of each module in LIPNovo. The results show that the imputation module accounts for only **0.46%** of the total pipeline runtime.
>
> In **Table 5**, we also provide the training time comparison, which shows that our LIPNovo adds 8 minutes per training epoch compared to the baseline model. All experiments were conducted on a server equipped with:
>
> - CPU: Intel(R)Xeon(R)Gold 5418Y @ 2.00GHz (96 cores,AVX2)
>
> - GPU: NVIDIA GeForce RTX 4090 (24GB)
>
> - OS: Ubuntu 20.04.6 LTS
>
> **Table 4:** Inference time (ms) and proportion of each module in LIPNovo.
>
> | Module         | Spectrum Representation | Imputation | Peptide Prediction | Total  |
> | -------------- | ----------------------- | ---------- | ------------------ | ------ |
> | **Time (ms)**  | 6.03                    | **3.60**   | 770.10             | 779.73 |
> | **Proportion** | 0.77%                   | **0.46%**  | 98.77%             | 100%   |
>
> **Table 5:** Training time (minutes) for an epoch ($499,402$ samples with batch size=$32$) compared to the baseline model.
>
> | Method              | Time (minutes) |
> | ------------------- | -------------- |
> | Baseline (CasaNovo) | 55             |
> | **LIPNovo (Ours)**  | 63             |
>
>
>
> [3]. Deng et al. Scalable analysis of cell-type composition from single-cell transcriptomics using deep recurrent learning, Nature Methods, 2019.

---

### Official Review · Reviewer_Dr28 · 2025-03-13

**Overall Recommendation:** 4

**Summary:**

This paper proposes LIPNovo, which is devised to compensate for missing fragmentation information within observed spectra before executing the final peptide prediction.

**Claims And Evidence:**

The claims made in the submission are supported by clear and convincing evidence

**Essential References Not Discussed:**

No essential references not discussed

**Experimental Designs Or Analyses:**

I check the validity of the experiments.

**Methods And Evaluation Criteria:**

The proposed method is novel, and the evaluation is fair.

**Other Comments Or Suggestions:**

No other comments or suggestions.

**Other Strengths And Weaknesses:**

This paper addresses the key issue of missing fragments in de novo peptide sequencing, which can advance progress in this field.

**Questions For Authors:**

It is better to demonstrate the experimental results to see whether this method is orthogonal to HelixNovo, which is also a method for solving the missing fragment problem. Using the complementary spectra as the LIPNovo's input.

**Relation To Broader Scientific Literature:**

Introduce missing data imputation into the de novo peptide sequencing.

**Theoretical Claims:**

This paper has no theoretical proofs.

---

> ### Author Rebuttal · Authors · 2025-03-30
>
> We sincerely appreciate the reviewer for the recognition that the proposed method is novel, and the evaluation is fair, which addresses the key issue of missing fragments in de novo peptide sequencing and can advance progress in this field. We address the reviewer's question as follows.
>
> **Q: It is better to demonstrate the experimental results to see whether this method is orthogonal to HelixNovo. which is also a method for solving the missing fragment problem. Using the complementary spectra as the LIPNovo's input.**
>
> Thanks for the thoughtful suggestion. $\pi$-HelixNovo and LIPNovo actually **address different aspects of the missing fragmentation issue**.
>
> $\pi$-HelixNovo handles the case where at least one ion ($b$ or $y$) in a pair is present, using the **complementary spectrum**. Specifically, if the $b$ ion is observed but the paired $y$ ion is missing, $\pi$-HelixNovo can recover the missing $y$ ion by subtracting the $b$ ion's mass from the precursor mass, which generates a complementary spectrum. However, $\pi$-HelixNovo cannot handle cases where a pair of $b$ and $y$ ions are all missing, which is how the missing ratio is calculated according to [2].
>
> **In contrast, LIPNovo is designed to impute all types of missing scenarios.** This is achieved by leveraging an imputation module that predicts the theoretical spectrum in a latent space, thus it effectively addresses problems that $\pi$-HelixNovo cannot resolve. This has been further evaluated through ablation experiments (as shown in **Table 3**). Specifically, "Baseline+ Complementary Spectrum" improves peptide precision by 0.8%. In contrast, "Baseline + our Imputation mechanism" achieves a **+4.0%** increase. Moreover, "Baseline + Imputation + Complementary Spectrum" further leads to 1.3% increase, because the complementary spectrum can provide additional information for the imputation process.
>
> **Table 3:** Experiments regarding the complementary spectrum used in $\pi$-HelixNovo.
>
> | Method                                                       | Peptide Precision | Peptide AUC  | Amnio-acid Precision | Amnio-acid Recall |
> | ------------------------------------------------------------ | ----------------- | ------------ | -------------------- | ----------------- |
> | Baseline (CasaNovo)                                          | 0.529             | 0.493        | 0.741                | 0.740             |
> | Baseline + **Complementary Spectrum**                        | 0.537(+0.9%)      | 0.500(+0.7%) | 0.755(+1.4%)         | 0.755(+1.5%)      |
> | Baseline + **Imputation Module**                             | 0.569(+4.0%)      | 0.536(+4.3%) | 0.782(+4.1%)         | 0.782(+4.2%)      |
> | Baseline + **Imputation Module** + **Complementary Spectrum** | 0.582(+5.3%)      | 0.547(+5.4%) | 0.797(+5.6%)         | 0.797(+5.7%)      |
>
> [2]. Zhou et al. NovoBench: Benchmarking Deep Learning-based De Novo Peptide Sequencing Methods in Proteomics, NeurIPS 2024 D&B track.

---

> > ### Comment · Reviewer_Dr28 · 2025-04-03
> >
> > Thank you for the author's response. Since my original score was positive, I will maintain it.

---

> > > ### Author Response · Authors · 2025-04-07
> > >
> > > Thank you for your reply and your time and effort in providing a thoughtful review of our work.

---

### Official Review · Reviewer_7ddk · 2025-03-20

**Overall Recommendation:** 3

**Summary:**

This paper presents a novel computational paradigm called **LIPNovo** for **de novo peptide sequencing**, addressing the problem of missing fragmentation information commonly encountered in mass spectrometry data. Unlike existing methods that rely on incomplete spectra, LIPNovo performs latent space imputation before prediction. The method formulates the imputation process as a set prediction problem, introducing a set of learnable peak queries to generate latent representations of theoretical peaks. Optimal bipartite matching is employed to align these predictions with ground truths, and an imputation loss function is designed to guide the process. Experiments conducted on the **Nine-species, Seven-species, and HC-PT datasets** demonstrate that LIPNovo significantly outperforms state-of-the-art methods across various metrics, including amino acid-level, peptide-level, and PTM-level performance. The proposed model achieves substantial improvements over the competitive baseline CasaNovo, confirming the effectiveness of the imputation mechanism. Ablation studies and sensitivity analyses further validate the contributions of each module. LIPNovo offers a powerful new approach to enhance peptide sequencing accuracy and robustness.

**Claims And Evidence:**

Yes

**Essential References Not Discussed:**

N/A.

**Experimental Designs Or Analyses:**

Yes. The method of this paper has been verified through experiments.

**Methods And Evaluation Criteria:**

Yes

**Other Comments Or Suggestions:**

Please see above.

**Other Strengths And Weaknesses:**

### **Advantages**
1. **Innovation**:
- It is novel to propose to fill in missing fragment information in the latent space instead of directly operating on the original mass spectrometry data.
- Bipartite matching is introduced into mass spectrometry data analysis, and the matching strategy in the field of target detection is borrowed to effectively solve the problem of predictive alignment of variable-length theoretical peaks.

2. **Technical Contribution**:
- An end-to-end filling-prediction framework is designed, and the filling process is guided by the latent representation of the theoretical spectrum, which significantly improves the robustness of the model to incomplete mass spectrometry data.
- The positive correlation between filling quality and sequencing performance is verified experimentally, and the necessity of each module is proved by ablation experiments and parameter analysis.

3. **Experimental Results**:
- On three mainstream data sets (Nine-species, Seven-species, HC-PT), LIPNovo has achieved SOTA in amino acid, peptide and PTM recognition, especially in the case of high fragment missing rate.
- The generalization and efficiency of the method were further verified by cross-validation and comparison with GraphNovo.

---

### **Improvement suggestions**
1. **Limitations of theoretical spectrum generation**:
- The theoretical spectrum relies on the ideal fragments of the target peptide (such as b/y ions), but in real scenarios, unknown peptides or complex PTMs may cause deviations in the theoretical spectrum. It is recommended to supplement the discussion on how to alleviate such problems.

2. **Computational efficiency analysis**:
- Filling modules may increase inference time, but the article does not mention the comparison of computational overhead (such as inference speed/memory usage with the baseline model). It is recommended to supplement relevant analysis to evaluate the practicality of the method.

3. **Improvement direction for extreme missing scenarios**:
- Experiments show that the performance improvement is limited when the fragment missing rate is >70%. It is recommended to further explore possible solutions (such as cross-sample learning or introducing auxiliary information) in the discussion section and design targeted experiments.

**Questions For Authors:**

N/A.

**Relation To Broader Scientific Literature:**

This paper presents a novel computational paradigm called **LIPNovo** for **de novo peptide sequencing**, addressing the problem of missing fragmentation information commonly encountered in mass spectrometry data

**Theoretical Claims:**

The method of this paper has been verified through experiments. No theory is proposed that requires additional proof.

---

> ### Author Rebuttal · Authors · 2025-03-30
>
> Thanks for the positive feedback on our work's novelty and the recognition that our method significantly improves the robustness of the model, achieves SOTA performance, and demonstrates generalization and efficiency. Our responses to the concerns are as follows.
>
> **Q1: Limitations of theoretical spectrum generation.**
>
> We make some clarifications about the theoretical spectrum generation.
>
> In our work, the theoretical spectra are computed from ground truth peptide sequences in the training set, where PTMs are explicitly considered. This ensures that peptides with different PTMs yield distinct theoretical spectra, and the mass deviations introduced by modifications are properly accounted for during training.
>
> Importantly, theoretical spectra are only used during training. Our method does **not require theoretical spectrum computation during inference**.
>
> LIPNovo is trained to map incomplete observed spectra to their corresponding theoretical spectrum representations. This acts as a **signal enhancement mechanism**, helping to mitigate deviations between observed and theoretical spectra and clarify spectral patterns, ultimately improving peptide sequence prediction.
>
> In more complex fragmentation scenarios (e.g. Electron Transfer Dissociation producing $c/z$ ions), the theoretical spectra can be extended to include these additional ion types. LIPNovo remains compatible with such scenarios **without requiring changes to its computational paradigm**, further demonstrating its flexibility.
>
> **Q2: Computational efficiency analysis.**
>
> Thanks for the valuable suggestion. We evaluate the computational overhead compared to the baseline in **Table 1**, which reports the inference time and GPU memory usage. The values are the mean across examples in the Nine-Species test set. The results show that LIPNovo increases the inference time by 14.82 ms (**+1.94%**) per example, and consumes an additional 98MB of GPU memory. All experiments were conducted on a server equipped with:
>
> - CPU:  Intel(R)Xeon(R)Gold 5418Y @ 2.00GHz (96 cores,AVX2)
>
> - GPU:  NVIDIA GeForce RTX 4090 (24GB)
>
> - OS: Ubuntu 20.04.6 LTS
>
> **Table 1:**  Comparison of computational overhead compared to baseline.
>
> |                     | Inference Time (ms) | **GPU Memory** (MB) |
> | ------------------- | ------------------- | ------------------- |
> | Baseline (CasaNovo) | 764.91              | 678                 |
> | **LIPNovo (Ours)**  | 779.73              | 776                 |
>
> **Q3: Improvement direction for extreme missing scenarios.**
>
> Thanks for the thoughtful suggestion. In **Lines 425-436**, we have discussed such limitations and introduced possible improvement directions. As indicated, one potential way is utilizing cross-sample information for missing peak imputation. To investigate this, we conduct a preliminary exploration to incorporate **cross-sample learning** into LIPNovo. Specifically, for each spectrum we search for the top one most similar spectrum (i.e., reference spectrum) from the training set with mass constraints, following [1]. Spectral similarity implies possible peptide similarity, where shared fragments can offer additional information for imputation. The reference spectrum is encoded by the spectrum encoder, and the resulting representation is fed into the imputation module to assist imputation.
>
> The results are presented in **Table 2**. As shown, under the high missing ratio range [0.7, 0.8), while the LIPNovo improves the baseline by only 0.86%, the simple extension **LIPNovo+Cross-Sample** achieves a **2.22%** improvement. On average, this extension enhances baseline performance by **8.38%** (*v.s.* 6.58% by LIPNovo).
>
> **Table 2:** Experiments on LIPNovo's improvement direction for extreme missing scenarios. Amnio-acid precision (%) is reported.
>
> | Missing Ratio Range       | [0,0.1) | [0.1,0.2) | [0.2, 0.3) | [0.3, 0.4) | [0.4, 0.5) | [0.5, 0.6) | [0.6, 0.7) | [0.7, 0.8) | [0.8, 0.9) | [0.9,1.0] | Mean   |
> | ------------------------- | ------- | --------- | ---------- | ---------- | ---------- | ---------- | ---------- | ---------- | ---------- | --------- | ------ |
> | Baseline (CasaNovo)       | 75.399  | 57.784    | 40.009     | 28.595     | 24.021     | 19.532     | 13.706     | 10.628     | 10.517     | 4.998     | 28.519 |
> | LIPNovo                   | 88.265  | 73.296    | 53.276     | 39.715     | 30.740     | 24.026     | 16.029     | 11.490     | 9.333      | 4.854     | 35.102 |
> | **LIPNovo+Cross-Sample**  | 88.946  | 74.296    | 55.983     | 41.622     | 33.831     | 26.047     | 17.712     | 12.849     | 11.069     | 6.624     | 36.898 |
> | $\Delta$ *w.r.t* Baseline | 13.547  | 16.512    | 15.974     | 13.027     | 9.810      | 6.515      | 4.006      | 2.221      | 0.552      | 1.626     | 8.379  |
> | $\Delta$ *w.r.t* LIPNovo  | 0.681   | 1.000     | 2.707      | 1.907      | 3.091      | 2.021      | 1.683      | 1.359      | 1.736      | 1.770     | 1.796  |
>
> [1]. Xia et al.  SearchNovo, ICLR 2025.

---

### Decision · Program_Chairs · 2025-05-01

**Decision:**

Accept (poster)

**Comment:**

This paper proposes LIPNovo, a framework aimed at addressing the (common) issue of missing fragmentation data in de novo peptide sequencing. The method joins set-prediction withg optimal bipartite matching to fill in missing spectral data within latent spaces, thus improving peptide prediction accuracy. The paper was met positively by the reviewers, who acknowledged its novelty and robustness of experimental validation. Strengths cited include the model's significant improvements over the CasaNovo baseline and its demonstration of generalisation and efficiency.

Review scores were 3, 4, 3. While one reviewer initially expressed concerns about the theoretical spectrum generation and computational efficiency, the authors effectively addressed these in the rebuttal. A detailed examination of cross-sample learning for handling extreme missing scenarios further validated the approach. Overall, reviewers highlighted the innovative nature of the imputation mechanism within latent spaces, marking it as a notable advancement in peptide sequencing methodologies.

Given the well-substantiated empirical results and the positive reassessment by the reviewers after the rebuttal, I recommend accepting this paper.